

# Survival rates of adult and juvenile gyrfalcons in Iceland: estimates and drivers

Frédéric Barraquand[1,2] and Ólafur K. Nielsen[3]

[1] Institute of Mathematics of Bordeaux, CNRS, Talence, France
[2] Integrative and Theoretical Ecology, Labex COTE, University of Bordeaux, Pessac, France
[3] Icelandic Institute of Natural History, Garðabær, Iceland

## ABSTRACT

Knowledge of survival rates and their potential covariation with environmental drivers, for both adults and juveniles, is paramount to forecast the population dynamics of long-lived animals. Long-lived bird and mammal populations are indeed very sensitive to change in survival rates, especially that of adults. Here we report the first survival estimates for the Icelandic gyrfalcon (*Falco rusticolus*) obtained by capture-mark-recapture methods. We use a mark-recapture-recovery model combining live and dead encounters into a unified analysis, in a Bayesian framework. Annual survival was estimated at 0.83 for adults and 0.40 for juveniles. Positive effects of main prey density on juvenile survival (5% increase in survival from min to max density) were possible though not likely. Weather effects on juvenile survival were even less likely. The variability in observed lifespan suggests that adult birds could suffer from human-induced alteration of survival rates.

## INTRODUCTION

Determining survival rates of animals is a prerequisite for building projections of their demographic trajectories. Many large birds are long-lived (*Sæther, 1989*), which creates a high sensitivity of the long-term growth rate and population persistence to adult survival rates (*Sæther & Bakke, 2000*). This makes the knowledge of adult survival rates key to predicting population conservation status in such long-lived birds (*Monzón & Friedenberg, 2018*). Juvenile survival rates are usually lower and more variable than adult survival rates in long-lived animals, since variation in vital rates critical to population growth is most often selected against (*Sæther & Bakke, 2000*; *Gaillard & Yoccoz, 2003*). This in turn makes the characterization of juvenile survival and its possible dependence on covariates important for understanding year-to-year population dynamics.

Adult and juvenile survival rates of gyrfalcons (*Falco rusticolus*) are currently unknown, despite an otherwise good knowledge of abundance and reproduction trends at the circumpolar scale (*Franke et al., 2020*), gene flow (*Johnson et al., 2007*; *Booms et al., 2011*), and diet as well as functional and numerical responses to changing prey density (*Nielsen, 1999*, *2011*). Indeed, this species population status is quantified with the number of territorial pairs and reproduction status (*Franke et al., 2020*, and refs. therein). This

Corresponding author
Frédéric Barraquand,
frederic.barraquand@u-bordeaux.fr

absence of knowledge of survival rates is slightly surprising, as the gyrfalcon is a large iconic bird of prey, the largest of all extant falcons.

Here, we estimate survival rates of adult and juvenile gyrfalcons in North-East (NE) Iceland with a Mark-Recapture-Recovery model, using both mark-recovery (dead recoveries) and capture-mark-recapture or -resighting data (live recoveries) collected over the period 1973–2019.

## MATERIALS AND METHODS

Gyrfalcons have been ringed in Iceland since 1939, with few captures before 1973. We therefore use data from 1816 capture histories from 1973 to 2019, including 293 recaptured individuals (either dead or alive), represented graphically in Fig. 1. Most have been ringed as nestlings, before 2011 only with engraved stainless steel rings and since then, also with engraved colour rings. In the 1990s, 24 territorial adults were trapped for ringing. EURING procedures were followed. As can be seen in Fig. 1, most recaptures are dead recoveries (only 75 re-sightings or live recaptures have occurred, with 246 dead recoveries; note that some individuals have had multiple recaptures). There have been a few more live recoveries in early and recent years, which is especially visible for young adults. We excluded data from 103 juvenile and adult birds that were released into the wild after rehabilitation and 5 nestlings that were marked with improperly sized rings. Gyrfalcon monitoring is done by ÓKN as project 2862 at the Icelandic Institute of Natural History (IINH).

The capture-recapture model used here is a multistate Mark-Recapture-Recovery model, following a long tradition of combining information of the various capture processes (live capture, dead recovery, resighting) to provide more robust estimates of survival probabilities (*Lebreton et al., 1995*; *Catchpole et al., 1998*; *Lebreton, Almeras & Pradel, 1999*). Our formulation is inspired by *Kéry & Schaub (2011)* but has a slightly different model structure. The model is formulated as a state-space model, more specifically as a Hidden Markov Model (see e.g. *Kéry & Schaub, 2011*; *McClintock et al., 2020*, for pedagogical expositions). It comprises therefore both a hidden state process, which models how the true states of bird individuals transition between years (transition matrix $\Gamma$), and an observation process which models how the true, hidden individual state of a bird translates into noisy human observations of that bird individual (observation matrix **O**). We develop the specifics of the model below.

We consider three observed states (denoted here $L$: seen or captured alive; $D$: recovered dead; $U$: neither seen, captured nor recovered) and three hidden states (1: alive; 2: recently dead; 3: long dead). In hidden state 1, live individuals can be observed, either through re-sighting or recapture, both of which are represented by observed state $L$, with probability $p$. State 1 individuals can also remain unseen (observed state $U$) with probability $1 - p$. In hidden state 2, birds are recently dead but exist as carcasses, which is made possible by the cold climate of Iceland. State 2 is pivotal to our model since dead recoveries (observed state $D$) can only occur when the individual is in process state 2, occurring yearly with probability $r$. State 2 individuals can also be unobserved ($U$ state) with probability $1 - r$. In hidden state 3, dead birds are no longer present in observable
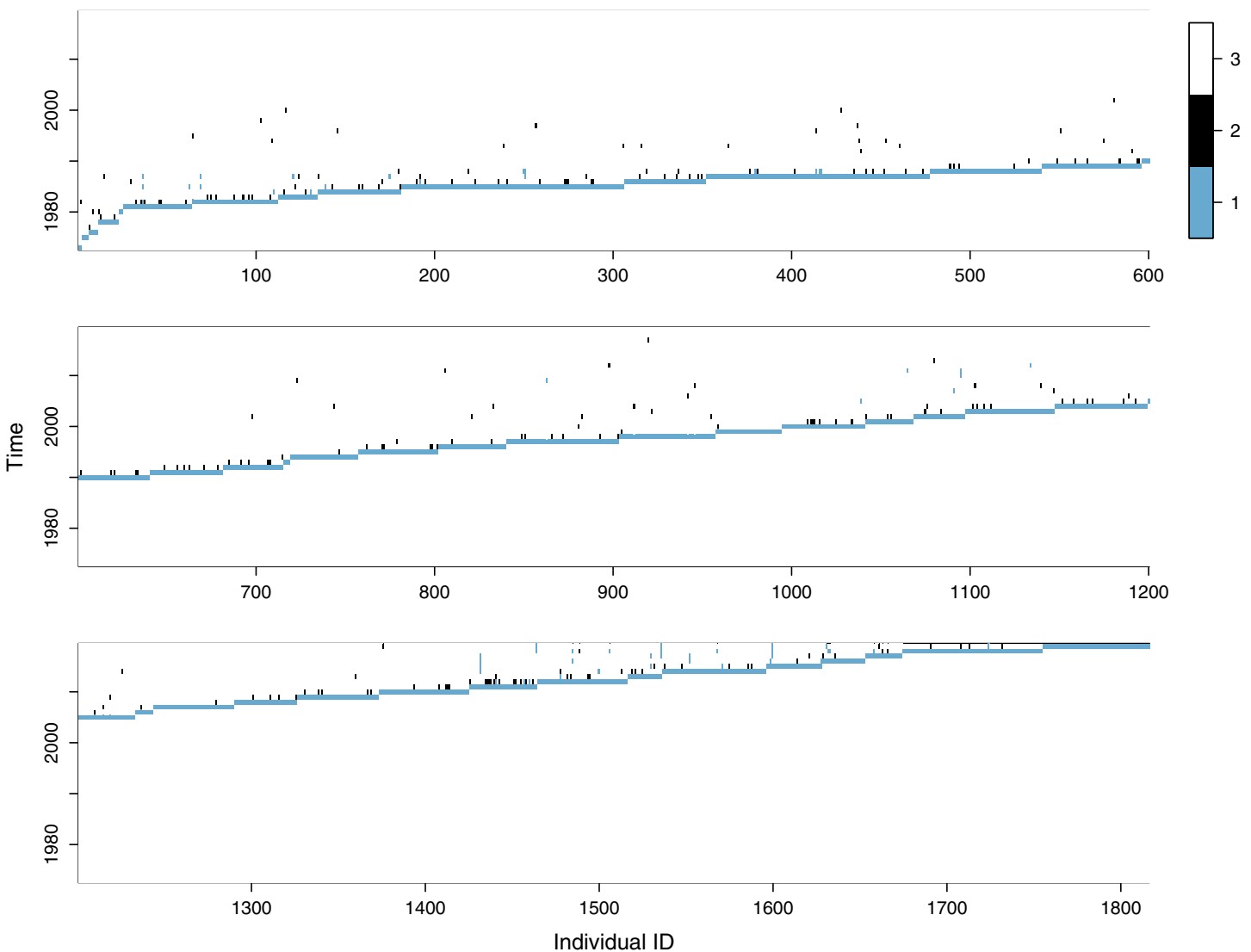

**Figure 1** Observed capture histories of gyrfalcons (*Falco rusticolus*) ringed in Iceland (1973–2019), with the state sequence for individuals ringed (1-blue: live capture or resighting; 2-black: dead recovery; 3-white: neither seen nor recovered).

form, and therefore state 3 gives observed state $U$ with probability 1. These relationships are encoded in the observation submodel, given by the observation probability matrix

$$
\mathbf{O}(t) = \begin{array}{c} \\ \text{Observed} \end{array} \begin{array}{c} \\ L \\ D \\ U \end{array} \begin{pmatrix} p & 0 & 0 \\ 0 & r & 0 \\ 1-p & 1-r & 1 \end{pmatrix} \tag{1}
$$

which quantifies the so-called *emission distribution* of the hidden Markov model.

Transition between the hidden states from one year to the next (using the calendar year) is modelled as a Markov chain, using the transition matrix of the hidden state submodel. The hidden state Markov chain is given by the following transition matrix

$$
\Gamma(t) = \begin{array}{c} \\ \\ \text{From state} \end{array}
\begin{array}{c}
\quad\; \text{To state} \\
\begin{array}{ccc} 1 & 2 & 3 \end{array} \\
\begin{array}{c} 1 \\ 2 \\ 3 \end{array}
\left( \begin{array}{ccc}
s_i(t) & 1 - s_i(t) & 0 \\
0 & \eta & 1 - \eta \\
0 & 0 & 1
\end{array} \right)
\end{array}
\tag{2}
$$

where we have highlighted the temporal dependence of $s_i$ since it can depend on covariates.

Quickly summarized, live birds survive from one year to the next with probability $s_1$ for juveniles and $s_2$ for adults, while carcasses may survive with probability $\eta$ from one year to the next (a probability that is typically very low). Our models included two stages, adult and juvenile, and considered that an individual is becoming an adult after 2 years of age (i.e., birds of age 3 and above are considered adults from the perspective of survival estimation).

Adult survival rates in these birds are unlikely to react to covariates since adults can rely for their own survival on a varied diet (*Nielsen & Cade, 1990b*). Moreover, as adults stay on their territories all through the year, they should have intimate knowledge of prey availability and distribution, as well as access to winter roost sites (*Nielsen & Cade, 1990a*). However, inexperienced juvenile birds might have a survival rate that fluctuates over the years, depending on the abundance of the main prey (rock ptarmigan, *Lagopus muta*) and weather variables.

We therefore considered three different models for the juvenile survival rate:

- Model A with constant juvenile survival rate $s_1$
- Model B with $s_1(t) = \text{logistic}(\mu_{s_1} + \beta x_{1,t})$ with $x_{1,t}$ = ptarmigan abundance for year $t$
- Model C with $s_1(t) = \text{logistic}(\mu_{s_1} + \beta_1 x_{1,t} + \beta_2 x_{2,t} + \beta_3 x_{3,t})$ with $x_{2,t}$ = winter temperature and $x_{3,t}$ = winter log-precipitation for year $t$.

We additionally considered model S—a 'silly' model—which assumes that juveniles and adults do not differ in their survival rates, for the sole purpose of checking that model selection was performing correctly.

These models are motivated by previous studies showing effects of ptarmigan abundance on gyrfalcon dynamics (*Barraquand & Nielsen, 2018*) with very weak and uncertain weather effects. That said, previous studies were focused on reproductive-density dynamics and not directly on vital rates, which creates potential for more direct weather effects here, since we know that conditions in the first spring (hatching) and subsequent winter of young gyrfalcons are probably important to their survival (*Nielsen, 2011*).

Ptarmigan abundance was calculated as the spatially averaged mean ptarmigan density index of the year as in *Barraquand & Nielsen (2018)*. The two weather variables considered were temperature and log-precipitation, constructed monthly as in *Barraquand & Nielsen*

*(2018)* by averaging over weather stations in NE Iceland; the only difference here is that we further averaged these variables over the period from October to March, this period being thought to be critical for juvenile survival. The covariates were standardized to allow for comparison, their means and SDs are given in Table S1.

We used weakly informative priors (e.g., $\beta_i \sim \mathcal{N}(0, 1)$) for regression coefficients, uniformpriors $\mathcal{U}(0, 1)$ for live survival probabilities, and $\mathcal{U}(0, 0.5)$ for detection probabilities and $\eta$ that were known from preliminary trials to be much smaller. A second set of more informative priors (Beta priors) was considered in additional analyses (see code folder in *Barraquand & Nielsen, 2021*) for robustness. The empirical support for the models was evaluated using Bayes factors, using the R package `bridgesampling` (*Gronau, Singmann & Wagenmakers, 2017*). We also used PSIS-LOO (*Vehtari, Gelman & Gabry, 2017*) to compare model predictive abilities. For a guide to Bayesian model comparison, see *Hooten & Hobbs (2015)*.

Models have been fitted in Stan (*Carpenter et al., 2017*) within R (*R Core Team, 2017*; *Stan Development Team & others, 2018*) using the Forward algorithm for Hidden Markov Models (*McClintock et al., 2020*), which helps to increase speed (*Yackulic et al., 2020*) and convergence. Similar estimates have been found computing the full latent states of the model (i.e., probability densities for the state of each individual at each moment in time) using JAGS for the simplest model without covariates, but these converged poorly. We ran all models for 1,000 iterations after a warm-up of 1,000 iterations and 3 chains, convergence was considered fine as $0.999 < \hat{R} < 1.001$. Code and data can be found at https://github.com/fbarraquand/Gyrfalcon_CMR and *Barraquand & Nielsen (2021)*. For models B and C, data on gyrfalcon capture histories were truncated, excluding data before 1981 (i.e., birds marked before that date were excluded) because all our covariates were only measured post-1980. This removed only seven birds that were recaptured or recovered.

## RESULTS

Model A estimated a probability of juvenile survival of 0.396 with 95% credible interval [0.341, 0.452], and an adult survival probability of 0.830 [0.790, 0.867]. The probability of detection $p$ was 0.020 [0.013, 0.029] for live birds and 0.139 [0.124, 0.156] for dead birds (i.e., probability of recovery $r$). The probability for a carcass to survive in the field for 1 year to the next was extremely low, 0.008 [0.000, 0.032].

We estimated using Model B the effect of ptarmigan density on juvenile survival, reproduced in Fig. 2 with credible intervals. A different parameterization $s_1 = \text{logistic}(-\gamma (x_1 - \mu_x))$ with informative priors, assuming the slope of the regression $\gamma$ to be positive, tended to increase the mean survival (Fig. S1). Because of the low consistency of the latter parameterization with the previous Model A estimates, we favoured the parameterization of Model B presented in Fig. 2. From the lowest to the highest ptarmigan densities, the increase in $s_1$ is slightly above 5%.

With Model C, we could compare the relative effects of ptarmigan density and weather variables on juvenile survival. It is shown in Table S1 and Fig. S2 that there could be

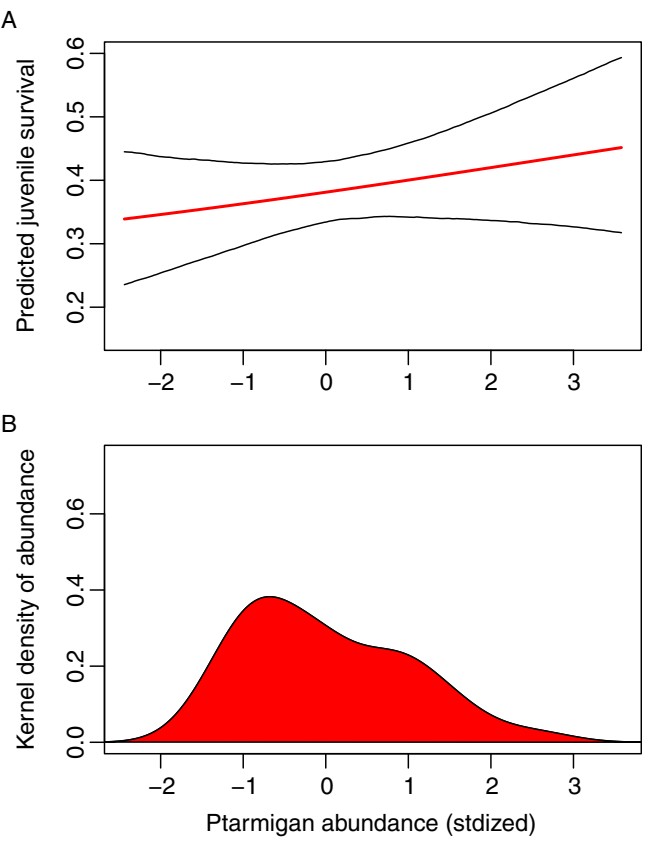

**Figure 2 Logistic regression of juvenile gyrfalcon survival ($s_1$).** (A) $s_1$ as a function of ptarmigan abundance $x_1$. (B) Corresponding distribution of $x_1$. Standardized ptarmigan abundance of −1.43 (observed min) and 2.58 (observed max) were approximately 2.5 and 12.5 individuals.km$^{-2}$, respectively.

positive effects of mean temperature (and more surprisingly, positive effects of precipitation).

We compared the models using Bayes factors (Table 1) which clearly favored model $A$ (without the covariates), suggesting that if the 'true model' were included within the set of models compared, that model would be model $A$. We also performed a comparison using cross-validation (PSIS-LOO) but the expected log predictive density did not differ substantially between models A, B and C (see Supporting Information), suggesting that these models have similar performances when compared in a predictive setup.

## DISCUSSION

The mean adult survival rate of 0.83 was in line with similar estimates for other raptors (*Newton, McGrady & Oli, 2016*; *Hernández-Matas, Real & Pradel, 2011*). Indeed, the regression of survival with body mass of *Newton, McGrady & Oli (2016)*'s meta-analysis, $Y = 0.437 + 0.052 \ln(x)$ with $x = 1{,}400$ g predicts 0.81 as annual survival rate. This average adult survival rate corresponds to a mean duration of the adult stage of 5.88 years, and thus an average lifespan given juvenile survival of 8 years. This suggests in turn substantial heterogeneity in lifespan between individuals (consistent with the geometric distribution

**Table 1 Bayes factors comparing models B and C, A and B as well as A and S.**

| Priors | $BF_{BC}$ | $BF_{AB}$ | $BF_{AS}$ |
|---|---|---|---|
| Uniform | 3.54 | 22.26 | $1.69 \times 10^{27}$ |
| Beta | 3.56 | 8.86 | $3.85 \times 10^{27}$ |

of lifespans assumed here), since the oldest individual lived up to 16 years (more than the 12 years suggested *Booms, Cade & Clum, 2020*) and three others to 15 and 14 years old. And it may well be that some individuals could live longer than that. Juvenile survival was estimated around 0.40, twice as low, which is in line with usual juvenile/adult survival ratios (*Newton, McGrady & Oli, 2016*).

In general, survival probability estimates such as those presented here or in *Newton, McGrady & Oli (2016)* are *apparent* survival estimates, affected by departures from the population. The particular setup of the study may bring these numbers closer to true survival estimates though, but this requires some explanation. Most birds have been ringed in NE Iceland (1642 in NE Iceland and 174 from other areas of Iceland), and clearly we would measure apparent survival if recovering only in NE Iceland. However, recoveries can occur outside NE Iceland. Indeed, for all birds ringed within the study area, 119 encounters are from outside the study area, and only 181 from within. Regarding the 23 encounters relating to birds ringed outside study area, 22 of the encounters took place outside the NE study area and only one inside. Because observations of ringed birds can come from all over Iceland, the probably of detection might not differ much between birds ringed in NE Iceland vs ringed in the rest of Iceland. Following this logic, survival probabilities should not be affected by this larger recapture area, so long as most juvenile birds stay within Iceland, as suggested by genetics data (*Johnson et al., 2007*) and recoveries of ringed birds. Patterns of movements within Iceland are largely due to juveniles: some degree of site fidelity is expected for adult gyrfalcons from territory data (*Nielsen, 1991*; *Booms et al., 2011*), but juveniles can often decide to settle in other regions.

To explain the substantial variation in observed lifespan between individuals mentioned above, we should therefore turn to other factors than emigration. Another likely contributor to heterogeneity in lifespan may be human-induced mortality. Indeed human-induced mortality has been found to substantially lower survival rates in other raptors (e.g., *León-Ortega et al., 2016*). Approximately 1 in 4 recovered and X-rayed gyrfalcons (18 birds out of 68 examined) have embedded shotgun pellets. Although the species has been protected since 1940, we can therefore expect a substantial direct or indirect mortality due to shooting.

From the lowest to the highest ptarmigan densities, the increase in juvenile survival rate is noticeable but barely above 5%, which does suggest a rather small effect of spatially averaged ptarmigan abundance. Model selection also indicated weak support for the model with the ptarmigan effect. This may sound surprising, given that ptarmigan is the gyrfalcon's main prey. However, we are working here with individual-level CMR data but with population-level covariates that exhibit temporal variation. All spatial variation is

therefore neglected: it might well be that some juveniles in ptarmigan-rich areas have better survival, we only measure here the effect of variation of ptarmigan abundance among years. Moreover, our ptarmigan index is for NE Iceland. Populations of ptarmigan within Iceland are not necessarily in synchrony; as juveniles from NE Iceland start dispersing during their first year of life, any spatial asynchrony should weaken the relationship between juvenile survival and our ptarmigan abundance index. Finally, this weak yet positive relationship between gyrfalcon juvenile survival and ptarmigan abundance might be due either to true survival variation among years or a slight emigration rate that is higher in ptarmigan-poor years; only movement studies may be able to resolve this ambiguity.

The survival rates and relationship with covariates estimated here represent what information one ought to incorporate into population models to forecast the future of the Icelandic gyrfalcon population. With the weak to nonexistent effects of ptarmigan abundance on juvenile survival estimated here, most effects of ptarmigan abundance on gyrfalcon dynamics that have been evidenced (*Nielsen, 2011*; *Barraquand & Nielsen, 2018*) likely occur due to the increased fecundity of gyrfalcons when ptarmigan are abundant. Weather effects on juvenile survival rates were not supported by model selection and the signs of the estimates were not always compatible with ecological logic. We therefore disregard effects of spatially and temporally averaged winter weather covariates on average juvenile survival, which will undoubtedly simplify future population-level models of gyrfalcon population dynamics. While it is possible that alternative ways to aggregate weather variables to the season and study area scales yield slightly different results, we view non-zero effects of averaged winter weather as rather unlikely because the weather is spatially variable and juveniles are, as stressed above, rather mobile. However, it should be noted that *local* winter weather—just as local prey abundance—may well impact the survival of individual juvenile gyrfalcons, depending on their locations and movements during the demanding winter period.

## ACKNOWLEDGEMENTS

We thank the Bird Ringing Office of the Icelandic Institute of Natural History for access to the gyrfalcon ringing data. Since 1994 the gyrfalcon studies have been part of ÓKN's job obligations at the Icelandic Institute of Natural History. ÓKN thanks all the people that have worked with him in the field over the years and helped with gyrfalcon ringing. FB thanks Hiroki Itò and Bob Carpenter who translated Marc Kéry and Michael Schaub's code into Stan, giving us a template to transition our JAGS code much more easily to Stan. We thank both referees for constructive criticism that prompted us to compare models more systematically.

### Funding

This work was funded by The Peregrine Fund, The National Geographic Society and The Icelandic Council of Science for Ólafur K. Nielsen and for Frédéric Barraquand by LabEx

COTE (ANR-10-LABX-45). The funders had no role in study design, data collection and analysis, decision to publish, or preparation of the manuscript.

## Grant Disclosures

The following grant information was disclosed by the authors:
The Peregrine Fund.
LabEx COTE: ANR-10-LABX-45.

## Competing Interests

The authors declare that they have no competing interests.

## Author Contributions

- Frédéric Barraquand conceived and designed the experiments, performed the experiments, analyzed the data, prepared figures and/or tables, authored or reviewed drafts of the paper, and approved the final draft.
- Ólafur K. Nielsen conceived and designed the experiments, performed the experiments, authored or reviewed drafts of the paper, did fieldwork for 35+ years, and approved the final draft.

## Field Study Permissions

The following information was supplied relating to field study approvals (i.e., approving body and any reference numbers):

Gyrfalcon monitoring has been authorized by Icelandic Institute of Natural History (IINH) since 1993, with project number 2862. Previously OKN had a permit from the Ministry of Education.

## Data Availability

All data and code is available at GitHub and Zenodo:

- https://github.com/fbarraquand/Gyrfalcon_CMR
- Frédéric Barraquand & Nielsen Ólafur Karl. (2021). Gyrfalcon survival estimation. Zenodo. https://doi.org/10.5281/zenodo.5459150.

## Supplemental Information

Supplemental information for this article can be found online at http://dx.doi.org/10.7717/peerj.12404#supplemental-information.

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
