# Peer review of "Survival rates of adult and juvenile gyrfalcons in Iceland: estimates and drivers"

_PeerJ, doi:10.7717/peerj.12404_

## Round 0.1 · original submission · Major Revisions

We have now received two in depth reviews which both enjoyed your manuscript and thought this was a good manuscript.

There are however several points that were raised, which I hope you can address in a revised version of you manuscript.

Finally, I just wanted to reiterate one of the points that was made by reviewer 2: "Furthermore, the analysis is entirely reproducible as the data and code are available online."

I am very happy to see this as I think that this is a crucial, but often underutilized part of science.

·

Basic reporting

I have some general comments on suggestions for alternative wording in section 4, but otherwise the manuscript is clear and well written. Given that neither author is from an English speaking country, I commend you on your writing.

The references are appropriate given the subject matter and I noted no obvious omissions. Perhaps more detail could be provided about ringing procedures, or simply refer readers to another paper for more details.

I thought Table 1 could be moved to supplemental (most of the parameter estimates and credible intervals are already cited in the text), and that the regression coefficients for Model C could be better displayed as a violin plot. Regrettably, I don't code in STAN so I wasn't able to thoroughly review the materials in GitHUB, but I went to the website and examined all materials. Everything is well documented.

The results are clear, and except for a few issues noted in general comments, I think that the authors did an outstanding job.

Experimental design

Materials and Methods: From figure 2 it seems that the covariates were all standardized, but you should probably state that clearly in the Methods. And for the regression parameters in Table 1 to make sense, you will need to inform the reader of the means and SD’s for each variable (this could be provided as supplemental information).

Validity of the findings

Results: It seems odd in an elegant Bayesian analysis to dismiss something as potentially important (e.g. winter temperatures, winter precipitation) because the 95% CI barely touches, or doesn’t quite touch, 0 (Fisher would be smiling in his grave).

Table 1: What does a SE mean in a Bayesian context? The SD is equivalent to a SE based on maximum likelihood estimation; the SE is just confusing. For that matter, this is just a table of STAN output that isn’t that helpful. You already include many of the parameter estimates and credible intervals in the text. I recommend making violin or rainfall plots of the posteriors for B1, B2, and B3, which will confirm that all 3 effects are mostly positive, and if all 3 are standardized, then the ptarmigan effect isn’t that much more pronounced than the temperature or precipitation effects.

Discussion (line 136-137): “This average adult survival rate corresponds to a mean duration of the adult stage of 5.88 years, and thus an average lifespan given survival of 8 years.” First, which adult survival rate do you mean by “this” because earlier in the paragraph you cite 0.83 as your empirical estimate (0.830 for Model A, but 0.834 in Table 1 based on Model C), but you also cite 0.81 based on the meta analysis of Newton et al.
Using your Table 1 value of 0.834 and calculating mean future lifespan for a newly banded adult, I get 1/-ln(S2) = 5.51 years, not 5.88 (using 0.830 gives 5.37 years).
And using the formula: 1/-ln(S1) + S1/-ln(S2) + S1/ln(S1) gives 2.78 years predicted mean survival for a newly banded juvenile (formula from Seber 1973 cited in Anderson, 1975, Population Ecology of the Mallard V). Perhaps you used a different formula to calculate mean expected lifespan, but because you haven’t indicated how you did this calculation, it’s difficult to interpret if it is appropriate or not.

Discussion (lines 138-148): Here you discuss evidence for individual heterogeneity, but your survival model for adults did not include an effect of individual heterogeneity (although Kery & Schaub consider such models) and therefore there is no evidence of survival heterogeneity based on your analysis.

Discussion (lines 141-144): These are apparent survival probabilities only if dead recovery data are limited to the mark-recapture study area. If the data include dead recoveries from gyrfalcons that dispersed off the study area (or could have been recovered if they dispersed off the study area), then these are true survival probabilities. You mention recoveries of dead juveniles from off the study area, so it seems that these are true survival probabilities, and the Discussion should be modified appropriately (see also lines 162-164, 175-178).

Additional comments

Minor comments (recommended wording changes, other comments and suggestions):

Abstract (line 10): “especially for adults” or “especially of adults”

Abstract (line 12): use “encounters” instead of “recaptures” because dead recoveries are not really recaptures.

Abstract (line 14): Reword as “with weak positive effects of prey density (ptarmigan), winter temperatures, and winter precipitation on juvenile survival”

Introduction, paragraph 1: Here you point out that adult survival is expected to show less annual variation than juvenile survival, based on Pfister (1998), but you don’t fit models that estimate juvenile and adult annual survival as random effects, which would allow you to test this idea. It seems odd to include this in the Introduction, but then not follow up on it (I could see citing it in the Methods as justification for why you don’t bother considering temporal variation for adults, but here in the Intro it seems to set up a hypothesis that is never tested).

Intro (line 19): Reword to: “for building population projection models.”

Introduction (line 24): delete “territorial” because it isn’t necessary. Delete “both” because it is confusing (also in line 27).

Introduction (line 28): important for what? Consider adding “important for understanding annual population dynamics”.

Introduction (line 29-30): I would consider omitting “a large iconic bird of prey – the largest falcon –“ because it breaks the flow of the sentence.

Intro (line 32): add “to changing prey abundance” to the end of the sentence.

Fig 1: The scale on the right side is unnecessary and confusing.

Material and Methods (line 44-45): I recommend rewording this sentence as “We excluded data from 103 adult birds that were released into the wild after rehabilitation and 5 nestlings that were marked with improper sized rings.”

Results: consider referring to the survival estimates as probabilities rather than rates, since they are measured on the interval scale and not instantaneously.

Line 49: “…estimates of survival”

Line 96: delete “dynamics” and just say “reproductive densities”.

Line 98: change “is” to “are” to match the plural “conditions” earlier in the sentence.

Figure 2A y-axis legend should say “Predicted” rather than “Predator”; understanding the biological magnitude of this effect would be easier if you provided information to help interpret the X axis; e.g. “Standardized ptarmigan abundance of -1 and 2 were approximately #.# and #.# individuals/km2, respectively.”

Discussion (line 183): replace “would likely enter predator demography through predator fecundity“ with “likely occur due to increased fecundity of gyrfalcons when ptarmigan are abundant.”

·

Basic reporting

1) Line 42 reads “As can be seen in Fig. 1, most recaptures are dead recoveries”. However, I find it rather difficult to extract this information from the figure. Thus, could you please add numbers here for clarification?
2) Although that might be my fault, I find Figure 1 rather difficult to interpret. On the y axis are plotted the years in which individuals have been observed (which usually was just one year), right? But what does the legend on the right mean? Could you please either revise the figure or add a sentence explaining it in more detail?
3) In my opinion, the paragraph in lines 77-82 should rather be placed at/towards the end of the methods section as you otherwise jump between presenting the model and explaining how you fit it.
4) Figure 2 (A): I suggest using the values from 0 to 1 as range of the y axis, or at least to ensure that the credibility intervals are fully displayed (which is not the case at the left boundary).
5) Figure 2 (B): The figure should show the whole range of abundance values on the x axis.
6) In my opinion, Table 1 could as well be placed in the Appendix. However, if you decide to keep it in the main text, it should display the same parameter estimates as the ones used for Figure 2 and presented in the text.
7) I believe the effects of the weather covariates (model C) should also be visualized such that their relevance for survival can easily be grasped, although the figures might rather be added to the Appendix.
8) Personally, I would find it quite helpful for the readers to include the transition matrix and the observation matrix in the main document as they nicely “summarize” the text and concentrate the parameters that are of interest. Why would you rather put them in the Appendix instead?

Experimental design

1) I am missing some sort of model comparison between model B and C (and possibly model A, although it is currently based on a somewhat larger data set, but could also be fitted on the smaller one). This would greatly help to assess the importance of the covariates for juvenile survival. Therefore, I suggest to perform some model selection first between the models A to C, and to then present the results of the chosen model (or at least either model B or C) in the main text. Currently, Figure 2A depends on different parameter estimates than those shown in Table 1, which can be rather confusing.
2) ll. 40-41: Do I understand it correctly that data on 1816 individuals has been used for analysis, of which only 293 have been recaptured and all others have been observed once? Could you please clarify how often individuals have been resighted or recaptured alive, and how many have been recaptured dead? I believe additional descriptive statistics on the data are necessary here.
3) In line 121 you present a different parameterization. The notation appears to be different to the one in line 91, why? And why has the covariate not been standardized in this model as well (cf. Figure A1 in SI)?

Validity of the findings

1) Regarding the effect of Ptarmigan abundance on the survival of juveniles (ll. 125-126): As the credible intervals are quite large, it is also possible that the abundance has no effect at all on survival. Therefore, I suggest formulating this result in a more cautious way (also in the abstract) and to point out the large uncertainty in the parameter estimate.
2) In line 166, you call the effect of Ptarmigan abundance on survival “moderate”. I would disagree – if there is any effect at all, it is rather small. Could you hence please rephrase this sentence.
3) The estimate of the probability of detection is very small (ll. 116-117). Is that realistic, i.e. are there similar results in other bird studies? How have the data been collected that the detection probability is so small, could you please elaborate on this?
4) In lines 183-188 you discuss the results of weather effects on juvenile survival. However, as there are huge uncertainties in these effects, I rather believe that one cannot conclude anything from these results (at most some minor indication for a possible positive correlation). Therefore, I suggest rephrasing this paragraph. In addition, one could think of missing covariates or interaction terms that should be considered in the model, which otherwise might mask the true relationship between survival and weather.

Additional comments

Overall, I enjoyed reading the manuscript, which is well written and clear structured. Furthermore, the analysis is entirely reproducible as the data and code are available online.
However, I do have some concerns regarding the presentation of the findings which should be improved upon before acceptance. In particular, I believe that performing model comparison/selection would greatly improve the understanding of the variables’ relevance for juvenile survival, while the uncertainty in these parameter estimates should be stressed when presenting the results.

---

## Round 0.2 · Minor Revisions

Thank you very much for submitting your revised manuscript!
As you will see from the reviewer comments, both reviewers very much appreciate your work on this revision very much.

Reviewer 2 has two more comments that they would like to see addressed. Could you please provide your comments and any changes you make in light of them in another revision?

Thanks very much!

·

Basic reporting

The paper is very well written (I have only a few small suggestions for rewording in the Additional Comments section).

Experimental design

The analysis is appropriate, and the Bayes' factors added to the revision make it easier to interpret the covariate effects as being mostly uninformative. I also think that moving the transition and observation matrices from the supplemental materials to the methods makes the paper easier to understand.

Validity of the findings

The authors provide the first published estimates of survival for Icelandic gyrfalcons. Based on added interpretation provided in the revised Discussion, it seems likely that these estimates are very close to true survival (i.e., unlikely to be strongly impacted by emigration).

Additional comments

Thank you for the careful revision. I believe your paper has been improved through your efforts, and will make an important contribution to the literature about raptor population dynamics.

I have highlighted 3 sections of the revised text where the wording is a bit awkward, and one correction for a figure legend. I have provided suggested revisions, but the authors are also welcome to provide their own revisions. Suggestions are in the attached pdf document.

·

Basic reporting

The manuscript is well written and clearly structured. The authors adequately addressed my comments from the previous review and, in particular, modified some figures, moved some material to the Appendix and reinserted some information into the main text. Overall, the reporting is greatly improved and hence, I do not have any further suggestions or comments.

Experimental design

As suggested, the authors now systematically compare their models using Bayes factors. This allows for a more thorough assessment of the importance of the covariates for juvenile survival, increasing the quality of the paper. However, I do have one last question:
You write that for models B and C, you exclude 7 capture histories from the data, right? But do you fit model A (and S) on the full data set? Because this would mean that you compare a model fitted to the full data set (model A) to a model fitted to a reduced data set (model B). As far as I know, all models need to be fitted to the exact same data set for model comparison with the same number of observations. Is this the case in your analysis (or maybe I just missed something)?

Validity of the findings

The authors adequately addressed all my concerns from the previous review and stress the uncertainty within their modelling results. There is only one final point regarding possible weather effects on juvenile survival rates, which I think is important to discuss.
You now write that “We therefore disregard those weather effects on juvenile survival. This will undoubtedly simplify future models of gyrfalcon population dynamics.” While I agree that in your current analysis, no weather effects on survival can be detected, I would not dismiss possible weather effects from all future analyses. I was wondering if maybe there is a ‘better’ or more accurate way of including those weather variables in the model. For example, I was wondering if it might make a difference if instead of averaging those variables, you would sum the log-precipitation, or would define a variable counting the number of days on which the temperature is below/above a certain threshold. I do not expect you to rerun your analysis, but I believe that these different possibilities for the variables to enter the model (different transformations, interaction terms, etc.) should be noted and might be explored in future studies – at least if/as there are reasons to suspect that weather conditions influence survival.

Additional comments

I believe the manuscript has greatly improved. I only have two more comments, which I think should be addressed before a possible publication of the paper.

---

## Round 0.3 · accepted · Accept

Thank you very much for addressing the remaining two comments from reviewer 2. In my opinion, you have sufficiently addressed their comments and I'm happy to recommend your study be published in PeerJ.